# Competing health risks associated with the COVID-19 pandemic and early response: A scoping review

Stefan Baral[1‡]*, Amrita Rao[1‡], Jean Olivier Twahirwa Rwema[1], Carrie Lyons[1], Muge Cevik[2], Anna E. Kågesten[3], Daouda Diouf[4], Annette H. Sohn[5], Refilwe Nancy Phaswana-Mafuya[6,7], Adeeba Kamarulzaman[8], Gregorio Millett[9], Julia L. Marcus[10], Sharmistha Mishra[11]

1 Department of Epidemiology, Johns Hopkins School of Public Health, Baltimore, Maryland, United States of America, 2 Division of Infection and Global Health Research, School of Medicine, University of St. Andrews, St. Andrews, Scotland, 3 Department of Global Public Health, Karolinska Institutet, Solna, Sweden, 4 Enda Santé, Dakar, Senegal, 5 TREAT Asia, amfAR, The Foundation for AIDS Research, Bangkok, Thailand, 6 South African Medical Research Council/University of Johannesburg Pan African Centre for Epidemics Research Extramural Unit, Johannesburg, South Africa, 7 Department of Environmental Health, Faculty of Health Sciences, University of Johannesburg, Johannesburg, South Africa, 8 Department of Medicine and Infectious Diseases, University of Malaya, Kuala Lumpur, Malaysia, 9 Public Policy Office, amfAR, Washington, District of Columbia, United States of America, 10 Department of Population Medicine, Harvard Medical School and Harvard Pilgrim Health Care Institute, Boston, MA, United States of America, 11 Li Ka Shing Knowledge Institute, St. Michael's Hospital, University of Toronto, Toronto, Canada

‡ SB and AR indicates co-first authorship to this work.
* sbaral@jhu.edu, sbaral@jhsph.edu

**Data Availability Statement:** All relevant data are within the paper and its Supporting Information files.

## Abstract

### Background

COVID-19 has rapidly emerged as a global public health threat with infections recorded in nearly every country. Responses to COVID-19 have varied in intensity and breadth, but generally have included domestic and international travel limitations, closure of non-essential businesses, and repurposing of health services. While these interventions have focused on testing, treatment, and mitigation of COVID-19, there have been reports of interruptions to diagnostic, prevention, and treatment services for other public health threats.

### Objectives

We conducted a scoping review to characterize the early impact of COVID-19 on HIV, tuberculosis, malaria, sexual and reproductive health, and malnutrition.

### Methods

A scoping literature review was completed using searches of PubMed and preprint servers (medRxiv/bioRxiv) from November 1st, 2019 to October 31st, 2020, using Medical Subject Headings (MeSH) terms related to SARS-CoV-2 or COVID-19 and HIV, tuberculosis, malaria, sexual and reproductive health, and malnutrition. Empiric studies reporting original data collection or mathematical models were included, and available data synthesized by region. Studies were excluded if they were not written in English.

**Funding:** Amrita Rao is supported in part by the National Institute of Mental Health [F31MH124458]. Carrie Lyons is supported by the National Institute of Mental Health [F31MH128079] and by the National Institute of Allergy and Infectious Diseases Johns Hopkins HIV Epidemiology and Prevention Sciences Training Program [T32AI102623-08]. Julia Marcus is supported in part by the National Institute of Allergy and Infectious Diseases [K01AI122853]. Sharmistha Mishra is supported by a Tier 2 Canada Research Chair in Mathematical Modeling. Refilwe Nancy Phaswana-Mafuya is supported by the South African Medical Research Council The funders had no role in study design, data collection and analysis, decision to publish, or preparation of the manuscript.

**Competing interests:** In terms of competing interests, Annette Sohn and Gregorio Millett report funding from ViiV Healthcare. Julia Marcus has consulted in the past for Kaiser Permanente Northern California on a research grant from Gilead Sciences. All other authors report no competing interests.

## Results

A total of 1604 published papers and 205 preprints were retrieved in the search. Overall, 8.0% (129/1604) of published studies and 10.2% (21/205) of preprints met the inclusion criteria and were included in this review: 7.3% (68/931) on HIV, 7.1% (24/339) on tuberculosis, 11.6% (26/224) on malaria, 7.8% (19/183) on sexual and reproductive health, and 9.8% (13/132) on malnutrition. Thematic results were similar across competing health risks, with substantial indirect effects of the COVID-19 pandemic and response on diagnostic, prevention, and treatment services for HIV, tuberculosis, malaria, sexual and reproductive health, and malnutrition.

## Discussion

COVID-19 emerged in the context of existing public health threats that result in millions of deaths every year. Thus, effectively responding to COVID-19 while minimizing the negative impacts of COVID-19 necessitates innovation and integration of existing programs that are often siloed across health systems. Inequities have been a consistent driver of existing health threats; COVID-19 has worsened disparities, reinforcing the need for programs that address structural risks. The data reviewed here suggest that effective strengthening of health systems should include investment and planning focused on ensuring the continuity of care for both rapidly emergent and existing public health threats.

## Introduction

The coronavirus disease 2019 (COVID-19) pandemic is among the most significant public health emergencies of international concern over the last hundred years causing substantial morbidity and mortality worldwide [1]. Public health responses to COVID-19 have varied in intensity, breadth, and duration, but in many countries have included domestic and international travel restrictions, stay-at-home orders and curfews, closure of non-essential businesses and schools, and repurposing of health services [2]. Although the goals of these interventions are to mitigate transmission of SARS-CoV-2 and ensure sufficient capacity for testing and treatment, such measures also have broader social, economic, and health impacts, including disruptions to routine public health programs and other clinical services [3–7]. Even when prevention or treatment services have remained uninterrupted, some people have been unwilling to seek care at healthcare facilities because of concerns about nosocomial SARS-CoV-2 acquisition risk or the misconception that services are only available for patients with COVID-19 [8].

Taken together, COVID-19 may have profound indirect and longer-term effects on broader health outcomes, including morbidity and mortality associated with other infectious and non-communicable diseases. Moreover, there may be increased risks of indirect health effects of the COVID-19 pandemic in low- and middle-income countries because of suboptimal healthcare resources and infrastructure. Within all countries, existing socioeconomic inequities, driven in part by structural racism, are likely to shape who is most affected, both directly and indirectly, by COVID-19. Marginalized groups that already experienced inadequate access to prevention, diagnostic, and treatment services, as well as a higher prevalence of other health conditions, may be most profoundly impacted by further interruptions to prevention, diagnostic, and treatment services during the pandemic response [9]. Understanding the indirect

effects of COVID-19 on health services, overall health outcomes, and equity is critical for planning and adapting public health responses to emerging infections, which need to maximize control of outbreaks while minimizing setbacks in other areas of health.

The objective of this review was to characterize the impact of early mitigation measures for the COVID-19 pandemic on health conditions that cause significant morbidity and mortality, including services and outcomes related to HIV infection, tuberculosis (TB), malaria, sexual and reproductive health (SRH), and malnutrition. To that end, we used a scoping review to summarize research findings and reporting from a range of different data sources and study types [10]. The results of the scoping review are synthesized, including implications for global health investments and policies that can mitigate the indirect effects of COVID-19 and future public health emergencies of international concern.

## Methods

We conducted a scoping review of published papers and preprints, and search strategies (**S1 Appendix**) were developed using Medical Subject Headings (MeSH) and key terms to focus on COVID-19 and its impact on one of the pre-specified key competing health risks: HIV, TB, malaria, SRH (contraceptives, abortion, pregnancy-related and newborn care), and malnutrition. Abstracts and full-text articles were reviewed using Covidence [11], an online systematic review management tool, and EndNote (version X8) [12].

For published papers, searches were implemented in PubMed. For preprints, we conducted a search of the COVID-19 Living Evidence Database, which includes published papers and preprints from both medRxiv and bioRxiv and is updated daily (https://zika.ispm.unibe.ch/assets/data/pub/search_beta/).

Articles were eligible for inclusion if the primary focus was the impact of COVID-19 on one of the five listed public health threats, and if they included empiric data or mathematical models on health or service outcomes. Articles were included if they were published or posted between November 2019 and October 2020 with the most recent search implemented on October 31, 2020. We excluded articles not written in English. Commentaries were screened for key themes to support interpretation of the measured and modeled data.

Titles and abstracts were reviewed by a single reviewer and selected for inclusion in this review based on the above criteria. The reviewers of each of the sections were as follows: HIV (AR), TB (CL), malaria (CL), SRH (JOTR), and malnutrition (JOTR). **Table 1** depicts the number of articles pulled, formally included based on the inclusion criteria, designated as commentaries, and excluded. Included articles were reviewed and an initial scan was completed for available empiric data or mathematical models and themes in terms of epidemiology of the burden or associated mortality of the condition and changes to service delivery. Some key indicators included changes in new infections and coverage of prevention and treatment services. Following this initial scan, calibration meetings were held in order to refine data charting processes, and final charting and synthesis were done independently.

## Results

### Outcomes of the scoping review

The systematic search identified a total of 1604 published papers and 205 preprints. For articles related to the impact of COVID-19 on the HIV pandemic, 7.3% (68/931) were deemed relevant and included in this review, along with 7.1% (24/339) for TB, 11.6% (26/224) for malaria, 7.8% (19/183) for SRH, and 9.8% (13/132) for malnutrition. The number of selected papers that were screened and eligible are reported in **Table 1**.

**Table 1. Number of papers pulled, included, marked as relevant commentaries, and excluded by competing health risk as part of the scoping review.**

|  | HIV | Malaria | Malnutrition | SRH | TB | TOTAL |
|---|---|---|---|---|---|---|
| PUBLISHED LITERATURE (PubMed) | | | | | | |
| **Retrieved** | 827 | 189 | 125 | 162 | 301 | 1604 |
| **Included** | 60 | 23 | 13 | 13 | 20 | 129 |
| **Commentaries** | 43 | 24 | 27 | 60 | 58 | 212 |
| **Excluded** | 724 | 142 | 85 | 89 | 223 | 1263 |
| PREPRINTS (COVID-19 Living Evidence database) | | | | | | |
| **Retrieved** | 104 | 35 | 7 | 21 | 38 | 205 |
| **Included** | 8 | 3 | 0 | 6 | 4 | 21 |
| **Commentaries** | 0 | 0 | 0 | 0 | 1 | 1 |
| **Excluded** | 96 | 32 | 7 | 15 | 33 | 183 |

SRH = Sexual and Reproductive Health; TB = Tuberculosis

## Summary of charted results

It was found that COVID-19 has been associated with reduced access to services, decreased diagnosis, and poorer health outcomes for HIV, TB, malaria, and SRH, and increases in malnutrition.

## COVID-19 impact on HIV services

Three main themes emerged from a review of the literature related to the potential impact of COVID-19 on HIV. The majority of papers included in this review described a destabilization of HIV service delivery and the negative impact of COVID-19 mitigation efforts on HIV testing, access to care, and viral suppression [13, 14]. Across geographic contexts, including in Australia, Indonesia, Italy, Kenya, and Uganda, fewer people living with HIV reported being able to attend clinic visits and access antiretroviral therapy (ART), resulting in a decline in the number of people estimated to be virally suppressed [15–19]. For example, in a global survey of men who have sex with men (MSM), close to 20% (218/1105) reported being unable to access their HIV provider during the pandemic and more than half reported being unable to refill their HIV prescription remotely (820/1254) [18]. In terms of HIV prevention, there was a decline in the number of people being tested and diagnosed [15, 20–22] and the amount of pre-exposure prophylaxis (PrEP) [23, 24] and post-exposure prophylaxis dispensed [18, 25, 26], although these declines may have been attributable in part to reductions in sexual activity [23, 24]. For example, one study of 847 gay and bisexual men in Australia found that about 42% discontinued PrEP use during the COVID-19 pandemic and that those discontinuing were less likely to report casual sexual partners [23]. COVID-19 responses have resulted in interruptions to the supply chains for the distribution of both ART and PrEP, and stock-outs of medications, as one study from Indonesia described [16].

Disparities were also identified in who was affected by interruptions to HIV prevention and treatment services. Specifically, existing socioeconomic inequities, including reduced access to health insurance and unstable housing, were associated with HIV service interruptions [14, 27]. In addition, key populations—including MSM, female sex workers, people who use drugs, and transgender populations—that depend on services from community-based organizations because of stigma within the health sector may have been particularly vulnerable to disruptions in outreach services caused by shelter-in-place mandates [28–30].

In some settings, there were reported adaptations in the implementation of HIV service delivery to mitigate interruptions, including adoption of telemedicine [19, 31–36]; home-

based HIV testing and self-testing [37–39]; home or mobile delivery of ART [34, 35, 40]; use of curbside pickup (i.e. pickup of supplies without stepping out of a vehicle) for condoms, lubricants, and medications [41]; and designation of surrogates such as peers to motivate and support ongoing treatment [19]. Adaptation of clinical services benefited from training of providers and approaches that promoted trust and took into consideration patients' needs and preferences [32]. Access to non-medical support, including cash transfers, reimbursement for the costs associated with accessing care [19], and housing and food supplementation support were key to promoting ongoing engagement in care [29, 39]. However, medical support and telemedicine strategies alone were unlikely to fully mitigate the poorer HIV outcomes observed; one study from a safety-net HIV clinic in San Francisco found that the odds of viral non-suppression were 31% higher after a shelter-in place mandate compared to before the mandate, even with telemedicine services [14]. Notably, most studies describing adaptations were in higher-income settings, though this may be reflective of a publication bias.

## COVID-19 impact on TB care cascade

Over the past several years, TB incidence and mortality have been steadily declining because of improvements in diagnosis, treatment, and prevention. The data available to date suggest that the COVID-19 pandemic may have had the unintended consequence of disrupting provision of TB services [43].

Reductions in timely diagnosis and treatment of new TB cases have resulted from COVID-19-related disruptions to access to healthcare services and availability of diagnostic capacity. Overwhelmed healthcare systems have often de-prioritized TB testing in laboratories and diverted these resources to COVID-19 testing [42]. In South Africa and Nigeria, for instance, GeneXpert machines and kits were prioritized for COVID-19 testing, leading to a drop of more than 50% in the median number of daily GeneXpert TB tests [43, 44]. Social distancing measures implemented in many countries disrupted patients' access to care, which impeded diagnosis, initiation of appropriate treatment, and follow-up. In Bangladesh, Kenya, Nigeria and Pakistan, the ability of residents in lower-income communities—which have a higher risk of TB—to seek healthcare for TB services has been reduced during COVID-19-related restrictions [45]. Missed diagnoses increase opportunities for transmission, while worsened treatment outcomes increase the risk of TB-related morbidity and mortality.

There have also been documented interruptions to services for people diagnosed with TB [46, 47]. For instance, in early 2020, there had been a substantial reduction in TB notifications in China, India, Japan, Nigeria, the Philippines, Sierra Leone, South Korea, and the United States compared with the same period in previous years [48–57]. Specifically, there was a decline of more than 50% in TB notifications in China in 2020 compared to 2015–2019 [48]. Furthermore, in Nigeria and South Korea, there was a one-third decrease in the number of active TB notifications in 2020 compared with prior years. As restrictive measures were lifted and COVID-19 rates declined, most of these settings reported an increase in TB notification rates [48–57]. In addition, reduced access to healthcare services and re-deployment of the TB workforce for the management of COVID-19 [58, 59] have created conditions for low adherence to treatment, which might also contribute to ongoing transmission and the emergence and spread of drug-resistant TB. During COVID-19-related restrictions in China, patient treatment completion and screening for drug-resistant TB among new and high-risk TB patients declined by approximately 20% [48].

At a broader level, COVID-19 prevention and mitigation measures have increased poverty and undernutrition, which are major risk factors for the acquisition and active conversion of TB. In India, an estimated 140 million people have lost their jobs during COVID-19, which

may indirectly exacerbate risk of TB [60]. In Brazil, the regions hardest hit by COVID-19 largely overlap with the regions where higher TB rates are observed [61]. With increased poverty and undernutrition, TB cases may surge among these disadvantaged communities. Furthermore, as regular services continue to be disrupted, routine TB immunization programs have been affected, such as in Pakistan, where an over 40% decline in BCG vaccinations has been reported [62].

Multiple modelling studies have estimated that COVID-19-related disruptions and fragmentation of TB services could result in an increase in TB incidence and mortality [63, 64]. The Stop TB Partnership reported that without countermeasures to maintain TB services, a 3-month period of COVID-19 restrictions followed by a 10-month recovery period could lead to an additional 6.3 million cases of TB by 2025 and an additional 1.4 million TB-related deaths in low- and middle-income countries [65]. Another modelling study estimated that over the next 5 years, these deaths could increase by up to 20% [66]. This emphasizes that the adverse effects of short-term disruptions will need to be addressed through "catch-up" TB case detection and treatment programs [67]. Critical efforts to mitigate impacts on TB control could include integration of TB and COVID-19 services for infection control, contact tracing, community-based care, surveillance, and monitoring. Innovative ways to deliver medicines and collect specimens for follow-up TB testing at home, and combine screening for TB and COVID-19, have already been demonstrated in South Africa [40, 68]. The socioeconomic inequities driving both TB and COVID-19 highlight the need for all countries to invest in universal health coverage and ensure equitable access to services.

## COVID-19 impact on malaria services

Studies have collectively demonstrated challenges in maintaining malaria prevention and control efforts in the context of COVID-19. In a study of 106 countries, 73% of malaria programs reported disruptions to service delivery, of which 19% reported high or very high levels of disruptions, potentially leading to increased morbidity and mortality [69, 70]. A resurgence of malaria due to COVID-19 may occur overall, and especially among vulnerable young children and pregnant women [71–73]. Of particular concern is the disruption of prevention efforts, including routine distribution of long-lasting insecticide-treated nets, seasonal malaria chemoprevention, and indoor residual spraying of insecticide [71, 74]. A modeling study suggested that the greatest impact on malaria burden could result from interruption of planned bed net campaigns, predicting 36% more deaths over 5 years in high-burden settings than would have occurred without COVID-19-related disruptions [66]. Another mathematical model suggested that COVID-19-related disruptions to malaria chemoprevention efforts and distribution of insecticide-treated nets in sub-Saharan Africa may have contributed to a doubling of malaria-related mortality in 2020 [75].

Malaria diagnoses during the COVID-19 outbreak have decreased, with a reduction in the notification rate as high as 62% in some settings [76, 77]. This reduction in diagnoses may be due to several factors, including reductions in health-seeking behavior related to malaria, as individuals may be reluctant to visit health facilities due to COVID-19 [71, 76]. Conversely, healthcare providers previously focused on delivering malaria care may have been reassigned to work on COVID-19, therefore limiting available malaria diagnostic services. COVID-19 and malaria have overlapping symptoms, including fever, difficulty breathing, headaches, and body pain, and there may be misdiagnosis of these infections in the context of limited laboratory testing [78, 79]. Furthermore, delays in reporting of malaria testing and confirmed cases have been observed, possibly due to disruptions in surveillance reporting structures [76]. Overall, undetected malaria infections as a result of the focus on COVID-19 testing threaten malaria control efforts [80].

Chloroquine (CQ) and its derivative, hydroxychloroquine (HCQ), are established prophylactic and clinical treatments for malaria and widely used in endemic areas. Early in the COVID-19 pandemic, these antimalarials were considered for potential treatment of COVID-19, and the U.S. FDA issued a temporary emergency use authorization for the use of HCQ for treatment of COVID-19, which was then rescinded in June 2020 [81]. Across settings, there was a documented increase in prescriptions of antimalarials, including an 80-fold increase in HCQ prescriptions in the U.S [81, 82]. Indiscriminate and widespread prophylactic and therapeutic use of CQ and HCQ for COVID-19 may complicate malaria prevention and control through several mechanisms. Alongside the increase in demand, shortages in the immediate supply may reduce their availability for use in malaria prevention and control, especially in low- and middle-income settings, which rely on international supply chains [83, 84]. Importantly, resistance to CQ and HCQ has previously emerged, and further indiscriminate use due to COVID-19 may drive Plasmodium resistance in malaria-endemic areas and threaten malaria control [73, 80, 82, 85].

## COVID-19 impact on sexual and reproductive health services

In terms of the impact of COVID-19 on SRH, we focused on early disruptions in selected essential services, including contraceptives, abortion, and pregnancy-related and newborn care [86]. Other SRH services and outcomes (e.g., screening, prevention, and care related to sexually transmitted infections, sexual violence, and reproductive cancers) were beyond the scope of our review.

Early during the COVID-19 outbreak, Roberton et al. [87] estimated that a 10–52% drop in service coverage would result in 12,000–57,000 additional maternal deaths over a 6-month period in low- and middle-income countries. Similarly, Riley et al. predicted that a 10% decline in SRH services would add 15.4 million unintended pregnancies, 3.3 million unsafe abortions, and 28,000 additional maternal deaths on a yearly basis [88]. Studies conducted during the initial phases of the pandemic outbreak and response further highlight SRH-related service disruptions, including 60 million fewer contraceptive users. These numbers are highest in Sub-Saharan Africa, Latin America, and the Caribbean, where the prevalence of provider-administered methods requiring face-to-face contact (such as injectable contraception) is the highest [89, 90].

Social distancing measures have resulted in interruptions to commodity production, supply chain delays, and clinic closures, resulting in commodity shortages [91]. In India, Marie Stopes International reported serving 1.3 million fewer women with contraceptive and abortion services than expected [92]. At the time the current search was conducted, research studies had documented mixed results in terms of the impact of COVID-19 on SRH service delivery outcomes [93–95]. Decreases in access to and use of contraceptives, antenatal care, safe abortion, and institutional delivery have been documented across different health systems and income contexts, including Kenya [96, 97], Ethiopia [98], Turkey [99], Italy [100], UK [101] and the US [102–104]. Notably, a recent large-scale prospective observational study in Nepal found a 52% decrease in institutional births coupled with increased neonatal mortality rates and poor intrapartum care during COVID-19-related restrictions [105].

Frontline maternal health workers have described changed care practices globally, such as relocation of human resources to the COVID-19 response, reduced face-to-face consultations, visitor bans (including for partners), and shorter post-delivery stays for mothers and infants [106–112]. Several adaptations have been introduced to mitigate the effects of these health systems challenges [113], including the Kenyan"Wheels for life initiative" to provide free transportation services to pregnant women during curfew hours [96],"click and collect" access to

contraceptives [114], and increased transition to telemedicine [110, 113–116]. In particular, newly imposed abortion restrictions in a number of European countries [117] and the US [102, 103, 118] during the initial COVID-19 outbreak have created an increased demand for medical abortion via telemedicine in several countries [117, 119–122]. This has been authorized in the UK in order to ensure equity and continuity or abortion care during COVID-19 [117]. However, technology requirements and legal restrictions on abortions mean that many women who need these adapted services will not be able to access them [119, 122]. These programs and workarounds highlight a demand for self-care services that will likely persist or grow in the future [123].

## COVID-19 impact on nutrition services

The immediate effect of COVID-19 on nutrition has been an increase in the number of individuals facing food insecurity in low-, middle-, and high-income countries [124, 125]. Food insecurity appears to be related to disruption of food supply chains due to limited movements of people and goods between countries, which in turn caused a disruption of food markets, increased food waste, and inflation of food prices [126–129]. This disruption of markets was exacerbated by the economic fallout associated with COVID-19, resulting in millions of people losing their sources of income, particularly in low- and middle-income countries, where the majority of individuals work in the informal sector. Though food insecurity has affected individuals of all demographics, children and women of low socioeconomic status have been particularly affected by service interruptions due to COVID-19, and the effects may be long-lasting among these populations.

School closures have resulted in loss of access to healthy foods for millions of school-aged children and adolescents who relied on schools to access healthy meals [130, 131]. In addition, among children younger than five years of age, malnutrition has been projected to increase, resulting in substantial morbidity and mortality [132]. For instance, a modelling study focusing on 118 low- and middle-income countries (LMIC) estimated that the disruptions in health services and increase food insecurity due to COVID-19 could result in a 14% increase in the prevalence of malnutrition, translating to 6.7 million more children under five experiencing severe malnutrition [87]. The same study estimated that COVID-19 would be associated with more than 120,000 additional deaths among children under five because of increased malnutrition and other unmet child health needs [87]. Maternal mortality is also expected to increase as a result of increased food insecurity and reduced access to maternal health programs [132, 133]. Specifically, an additional 12,200 to 56,700 maternal deaths could occur as a result of disruption in maternal health and nutrition programs in LMIC [87].

Given the substantial negative effects on nutrition and the associated morbidity and mortality, several papers have lamented the lack of explicit nutrition programs in the COVID-19 response and called for integration of nutrition programs within COVID-19 mitigation strategies. Specific strategies to mitigate increased malnutrition could include population-level interventions to support the communities most vulnerable to malnutrition [134]. In Nepal, for example, specific interventions to offset risks associated with restrictive COVID-19 interventions included continuation of vitamin A supplementation and provision of deworming tablets to children, programs supporting breastfeeding and other complementary foods, distribution of fortified foods to pregnant women, and ensuring the continuity of other existing maternal and child programs [133].

In addition to food insecurity, there has been an increase in unhealthy eating habits since the start of COVID-19. A cross-sectional study among over 1000 adult participants in Poland found that 43% of participants reported eating more frequently and 50% reporting more

snacking. Furthermore, 30% of participants in the study reported weight gain since the initiation of COVID-19-related restrictions, while 15% of participants reported consuming more alcohol and 45% of smokers reported increased smoking frequency [135].

## Discussion

Across high-, middle-, and lower-income countries, COVID-19 has been associated with reduced access to services, decreased diagnoses, poorer health outcomes for HIV, TB, malaria, and SRH, as well as increases in malnutrition. The most affected populations appear to be communities already on the margins, including those with lower income, racial and ethnic minorities, and women, resulting in the amplification of existing health inequities. In 2015, all United Nations Member States adopted the 2030 Agenda for Sustainable Development with a focus on 17 Sustainable Development Goals (SDGs). To support health and wellbeing for all, the SDGs laid out ambitious plans for zero new HIV, malaria, and TB infections by 2030, and ambitious goals to address malnutrition and reproductive health. Increases in communicable diseases and malnutrition, worsened reproductive health outcomes, and widening inequities could collectively result in a reversal of global health gains in key indicators [136].

The indirect effects of the COVID-19 pandemic may force a reexamination of global health investments and policies. Specifically, it has been estimated that as much as 90% of countries have experienced declines in per capita income due to the COVID-19 pandemic and responses [137]. These decreases have prompted questions regarding the viability of the Sustainable Development Goals and whether they should reflect more achievable targets in the wake of programmatic disruptions due to COVID-19 [138]. In addition, as service disruptions and COVID-19-related restrictions are expected to disproportionately affect already-marginalized groups—such as adolescents, sexual and gender minority communities, people living with HIV, refugees and migrants, and people facing gender-based violence [88, 97, 139–141]—human-rights affirming, intersectional approaches for monitoring and addressing the indirect effects on programs are critical [18, 139]. Specific funding support to non-governmental organizations with strong connections to these communities may be able to overcome disruptions in health services during public health emergencies. Moreover, efforts to decriminalize marginalized populations should be prioritized to promote legal and economic opportunities, as well as access to health care. Finally, there have been calls for more resilient supply chains for medicine and food in low- and middle-income countries. This requires supporting local market chains and removing intellectual property barriers to and within those countries to avoid reliance on international food and medicine supply chains and strengthen their production and delivery of biofortified foods and pharmaceutical interventions [127]. The importance of local production of pharmaceutical products including vaccines has been highlighted throughout COVID-19 given the extreme inequities that have defined COVID-19 vaccine distribution, availability, and ultimately, coverage.

Despite differences of opinion in specific policy actions, there are a few areas where a broad consensus is emerging. Multilateral initiatives and commitments are more important than ever, and donors must redouble their efforts to invest in global health efforts rather than retrench to keep from losing decades of gains. COVID-19 has stressed the capacity of health systems because of vertical and siloed health infrastructure designed to respond to specific diseases. Integrated health systems can not only address a multiplicity of health issues, but also can support integrated surveillance, data systems, supply chain management, and delivery [142]. To inform these policy initiatives, there is a need for disease-specific approaches to shift towards studying communities of individuals and health systems.

The benefits of scientific discovery are not linear with respect to disease. For instance, scientific advances in HIV have benefited cancer and hepatitis research, and have served as a basis for COVID-19 vaccines [143, 144]. Similarly, these health conditions themselves are interrelated. For instance, TB is the leading cause of death for people living with HIV in sub-Saharan Africa [145]. However, disease-specific research has often failed to study and respond to the complexities of this reality. Before COVID-19, for example, it would have been unusual to conduct research on the impact of a respiratory virus infection on domestic violence, depression, and job security among women living with HIV who were diabetic. However, given anecdotal and media reports of these relationships, research should evolve accordingly to better inform the lived realities of the public. Cross-disciplinary research initiatives can characterize the direct and indirect impacts of COVID-19, including implementation research on syndemic-related health outcomes, effects of legal policies and increased policing (e.g., protections for marginalized populations), structural racism, and issues of food security and employment [146–149]. In addition, studies on optimizing resource allocation and supply chain management for therapeutics and vaccines are critical to avoid worsening of inequities during scale-up [150]. To respond to these needs, the WHO convened a Global Research Forum early in February 2020 to accelerate research on the immediate priorities of COVID-19 mitigation and treatment, with secondary aims to build up global research platforms and drive equitable access to diagnostics and therapeutics [116]. Moreover, there was to be intentional assessment of how public health strategies may impact a multitude of factors across physical and mental health, as well as social infrastructure, economies, and politics [151–153].

Mathematical models have played a significant role in COVID-19-related decision-making. As in previous outbreaks and pandemics, transmission dynamics and statistical modelling provided rapid estimates drawing on rapidly evolving information [154, 155]. Similar to the policy and research initiatives to date, infectious disease modelling has remained largely "vertical", or siloed by health threats, thus resembling and informing decision-making for vertical health services. As models are expected to continue to drive decision making, the next generation of pandemic preparedness models could integrate case projection for an emerging infectious disease, and disease-specific health care and public health needs, with adaptive strategies for a resilient health care system. In projecting how many acute care hospital beds might be needed to care for patients with severe COVID-19, integration of other conditions could inform strategies to manage the surges while also minimizing disruptions to other health care services. For example, if estimates of the reduction in contact rates required to decrease SARS-CoV-2 spread were integrated, a priori, decision-makers could better understand differential COVID-19 mitigation strategies alongside systems-level modelling to inform decision-making across health services [156]. Separate from variability in underlying mortality, health system infrastructure varies significantly across regions, including health and human services per population or in-hospital and intensive care beds per capita. Further integration of localized health infrastructure parameters with COVID-19 transmission models would also support localized decision making for optimal interventions [157–159].

There are several limitations of this scoping review. Given the breadth of information reviewed, including over 1800 peer-reviewed and preprint articles, we did not conduct a formal population-intervention-control-outcome systematic review. In addition, the search strategy was only implemented in PubMed, medrxiv, and biorxiv, which means we may have missed other articles that were only indexed in other databases. Moreover, there were several areas not covered in this review, including vaccine-preventable diseases [160], non-communicable diseases [4, 161], specific health effects among migrant communities [162], violence [163], and mental health [164]. The timing of this review includes a focus on 2020 representing the earlier phase of the pandemic and early mitigation efforts though a non-systematic

assessment in November, 2021 suggested that the trends reported were sustained. Preliminary data suggest effects across all these areas, including reports of increased domestic violence, decreases in childhood vaccinations, and increases in mortality and morbidity because of acute mental health stressors and substance use. In addition, publication bias may have affected the estimates reported here. Because we did not complete a quantitative meta-analysis, we were unable to assess the magnitude of publication bias. Finally, there may be limited generalizability of the indirect effects of COVID-19 across regions given significant variability in the underlying causes of morbidity and mortality and varying health systems and health infrastructure.

## Conclusions

The COVID-19 pandemic has exposed disparate risks and inequities by income, race and ethnicity, gender, and immigration status. The results of this scoping review demonstrate ways that the COVID-19 pandemic and response have impacted other diseases and essential services, risking decades of progress in outcomes associated with HIV, TB, malaria, sexual and reproductive health, and malnutrition. Given vaccine inequity, the places most affected by these conditions were also the least able to support the vaccination of their population—likely causing even greater morbidity and mortality. Many settings have adapted by developing programs to mitigate the indirect effects of COVID-19, but optimizing population-level health in the context of public health emergencies of international concern necessitates broader innovation in research, mathematical modeling, policy, and programs. Moreover, a cross-disciplinary research agenda for pandemic preparedness and response modelling offers an opportunity to examine optimal decision making for health care and public health systems by integrating counterfactuals ('what if' experiments) for one disease with those for other health conditions. COVID-19 responses should also include a rights-based approach that helps ensure equitable access to prevention, diagnostic, and treatment services for both COVID-19 and competing health risks. The redesign and strengthening of health systems must include the strengthening of public health systems, with adequate funding and planning to ensure continuity of contextually relevant health and social welfare programs that specifically address the needs of communities most socially and economically marginalized.

## Supporting information

**S1 Checklist. Preferred Reporting Items for Systematic reviews and Meta-Analyses extension for Scoping Reviews (PRISMA-ScR) checklist.**
(DOCX)

**S1 Data. Papers included in the scoping review by pre-identified competing health risks of COVID-19: HIV, tuberculosis, malaria, sexual and reproductive health, and nutrition.**
(XLSX)

**S1 Appendix. Search strategies for the scoping review using Medical Subject Headings (MeSH) and key terms to focus on COVID-19 and its impact on one of the pre-specified key competing health risks: HIV, TB, malaria, SRH (contraceptives, abortion, pregnancy-related and newborn care), and malnutrition.**
(DOCX)

## Author Contributions

**Conceptualization:** Stefan Baral, Amrita Rao, Daouda Diouf, Annette H. Sohn, Refilwe Nancy Phaswana-Mafuya, Adeeba Kamarulzaman, Gregorio Millett, Julia L. Marcus, Sharmistha Mishra.

**Data curation:** Stefan Baral, Amrita Rao, Jean Olivier Twahirwa Rwema, Carrie Lyons, Muge Cevik, Anna E. Kågesten.

**Formal analysis:** Amrita Rao, Jean Olivier Twahirwa Rwema, Carrie Lyons, Muge Cevik, Anna E. Kågesten.

**Investigation:** Stefan Baral, Amrita Rao, Daouda Diouf, Annette H. Sohn, Refilwe Nancy Phaswana-Mafuya, Adeeba Kamarulzaman, Gregorio Millett, Julia L. Marcus, Sharmistha Mishra.

**Methodology:** Stefan Baral, Muge Cevik, Anna E. Kågesten, Julia L. Marcus, Sharmistha Mishra.

**Project administration:** Amrita Rao.

**Writing – original draft:** Stefan Baral, Amrita Rao, Jean Olivier Twahirwa Rwema, Carrie Lyons, Muge Cevik, Anna E. Kågesten, Daouda Diouf, Annette H. Sohn, Refilwe Nancy Phaswana-Mafuya, Adeeba Kamarulzaman, Gregorio Millett, Julia L. Marcus, Sharmistha Mishra.

**Writing – review & editing:** Stefan Baral, Amrita Rao, Jean Olivier Twahirwa Rwema, Carrie Lyons, Muge Cevik, Anna E. Kågesten, Daouda Diouf, Annette H. Sohn, Refilwe Nancy Phaswana-Mafuya, Adeeba Kamarulzaman, Gregorio Millett, Julia L. Marcus, Sharmistha Mishra.

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
