## [Decision Letter · Decision Letter 0]

12 Nov 2021

PONE-D-21-08875

Competing Health Risks Associated with the COVID-19 Pandemic and Response: A Scoping Review

PLOS ONE

Dear Dr Baral,

My deepest apologies for the delay in decision.

There has been a difficulty in recruiting editors in the past period. In my experience this has been across the board and not related to the substance of your submission.

I have had a reviewer submit their review and have reviewed it myself as well. Wanting to move forward I will accept publication with one reviewer (after you kindly revisit the text with the reviewers).

I apologize again for the delay.

Best, Tammam Aloudat

We look forward to receiving your revised manuscript.

Kind regards,

Tammam Aloudat

Academic Editor

PLOS ONE

Journal Requirements:

7. We note the search of the relevant literature is limited to searches on PubMed. We do not feel that this constitutes a search that is comprehensive enough to collect all relevant evidence available from the literature, and as a result, we have concerns about the strength of the conclusions drawn from the pool of studies included in your synthesis. Therefore, please extend your search to include other relevant databases and include any additional papers in your review, if applicable. 

8. In your methods section, please report the inclusion and exclusion criteria used in your review.

Reviewers' comments:

Reviewer's Responses to Questions

**Comments to the Author**

1. Is the manuscript technically sound, and do the data support the conclusions?

Reviewer #1: Yes

2. Has the statistical analysis been performed appropriately and rigorously? 

Reviewer #1: N/A

3. Have the authors made all data underlying the findings in their manuscript fully available?

Reviewer #1: Yes

4. Is the manuscript presented in an intelligible fashion and written in standard English?

Reviewer #1: Yes

5. Review Comments to the Author

Reviewer #1: 1. Line 133 and 134 do not state clearly the period that the authors concentrated in their search for published literature and preprints. Just stating “through October 31st” is not enough and is, in fact, ambiguous

2. In lines 182 and 183, the authors make a statement about the decrease in dispensing of pre-exposure prophylaxis claiming it might be due to reduced sexual activity, this statement should probably be referenced or removed entirely with the authors sticking to the referenced materials since this is not the discussion section of the paper

3. It will also be good if the use some actual values from the quoted sources to show the decrease number of persons being tested and also accessing pre-exposure prophylaxis

4. Concerning lines 225 and 226, which period of the year are the authors referring to? Do they mean the whole of 2020? Nigeria, for instance, had an overall increase in TB case finding in 2020 compared to previous years. The authors should look at the statement and consider rephrasing it to reflect the period in 2020 they were referring to.

5. The opening statements of the discussion seems like something that would have fitted well as part of the background for this study. It sounds like a more catchy way of introducing this study.

6. PLOS authors have the option to publish the peer review history of their article (what does this mean?). If published, this will include your full peer review and any attached files.

Reviewer #1: **Yes: **Okoro Chika Augustus

---

## [Author Response · Author response to Decision Letter 0]

5 Dec 2021

We thank the reviewer for taking the time to provide a careful review of our paper. We have responded in detail, point-by-point below. 

Reviewer #1: 1. Line 133 and 134 do not state clearly the period that the authors concentrated in their search for published literature and preprints. Just stating “through October 31st” is not enough and is, in fact, ambiguous

Thank you for this point, and we agree that the way this is stated is ambiguous. We have revised this to define the search period more clearly (now lines 144-145). 

2. In lines 182 and 183, the authors make a statement about the decrease in dispensing of pre-exposure prophylaxis claiming it might be due to reduced sexual activity, this statement should probably be referenced or removed entirely with the authors sticking to the referenced materials since this is not the discussion section of the paper

Thank you for this point. We had noted this point about a decrease in sexual activity, in conjunction with the decline in PrEP results, as this was also a primary result observed and presented in the papers reporting declines in PrEP and PEP use. We have now more clearly referenced this statement and added an additional example to clarify the point:

“For example, one study of 847 gay and bisexual men in Australia found that about 42% discontinued PrEP use during the COVID-19 pandemic and that those discontinuing were less likely to report casual sexual partners [23].”

3. It will also be good if the use some actual values from the quoted sources to show the decrease number of persons being tested and also accessing pre-exposure prophylaxis

Thank you for this suggestion. We have now provided some specific values on the decrease in testing and PrEP use seen during the COVID-19 pandemic. 

4. Concerning lines 225 and 226, which period of the year are the authors referring to? Do they mean the whole of 2020? Nigeria, for instance, had an overall increase in TB case finding in 2020 compared to previous years. The authors should look at the statement and consider rephrasing it to reflect the period in 2020 they were referring to.

Thank you for this suggestion. We have revised this statement to clearly indicate which period of 2020 we are referring to here. The statement now reads: “For instance, in early 2020, there had been a substantial reduction in TB notifications in Sierra Leone, China, Nigeria, South Korea, India, the Philippines, Japan, and the U.S. compare do the same period in previous years [48-57].”

5. The opening statements of the discussion seems like something that would have fitted well as part of the background for this study. It sounds like a more catchy way of introducing this study.

We agree with your assessment and have changed the opening lines of the discussion to focus on summarizing the preliminary and overarching findings of the Review.

---

## [Decision Letter · Decision Letter 1]

3 Jun 2022

PONE-D-21-08875R1Competing Health Risks Associated with the COVID-19 Pandemic and Early Response: A Scoping ReviewPLOS ONE

Dear Dr. Baral,

Thank you for submitting your manuscript to PLOS ONE. After careful consideration, we feel that it has merit but does not fully meet PLOS ONE’s publication criteria as it currently stands. Therefore, we invite you to submit a revised version of the manuscript that addresses the points raised during the review process.

Thanks a lot for properly addressing the concerns of the reviewers in the first round of revision. However, additional important and relevant points were raised by the reviewer #2 in this round of peer review. Kindly provide a detailed point-by-point responses to these concerns.

Please note that PLOS ONE will consider scoping reviews for publication after peer review if the manuscript meets the guidelines outlined in: https://journals.plos.org/plosone/s/submission-guidelines#loc-systematic-reviews-and-meta-analyses

We look forward to receiving your revised manuscript.

Kind regards,

Malik Sallam, M.D., Ph.D.

Academic Editor

PLOS ONE

Journal Requirements:

Reviewers' comments:

Reviewer's Responses to Questions

**Comments to the Author**

1. If the authors have adequately addressed your comments raised in a previous round of review and you feel that this manuscript is now acceptable for publication, you may indicate that here to bypass the “Comments to the Author” section, enter your conflict of interest statement in the “Confidential to Editor” section, and submit your "Accept" recommendation.

Reviewer #1: All comments have been addressed

Reviewer #2: (No Response)

2. Is the manuscript technically sound, and do the data support the conclusions?

Reviewer #1: Yes

Reviewer #2: Yes

3. Has the statistical analysis been performed appropriately and rigorously? 

Reviewer #1: Yes

Reviewer #2: N/A

4. Have the authors made all data underlying the findings in their manuscript fully available?

Reviewer #1: Yes

Reviewer #2: Yes

5. Is the manuscript presented in an intelligible fashion and written in standard English?

Reviewer #1: Yes

Reviewer #2: Yes

6. Review Comments to the Author

Reviewer #1: This paper has been properly written. All the previous concerns i had, especially in the presentation of the results were addressed by the authors.

Reviewer #2: Abstract

-It is very confusing the figures got from their literature search when mentioned. 1604 published papers and 205 preprints met inclusion criteria, but only 132 and 21 were selected, respectively. They have to replace the verb “including” by “selecting”.

Introduction

-References are not properly used in some instances. Reference [2] is not related to Public Health responses but to estimating the success of HIV care in key population.

-Lines 123-124. Delete “rather than systematic review”. If they want to discuss this point, relocate it in Discussion section.

-Methods

-Line 130. The other reviewer has already pointed that this sentence is confusing or trifling. In my opinion, they should merge the first sentence and the following one, or simply delete the first one.

-Line 131. They must enter the MeSH acronym the first time it was used in the text, as they did it in Abstract.

-There is a contradictory information regarding period of inclusion. In Abstract, the period cited ranged from January 1st to October 31st, 2020, whereas in Methods (line 144) the period started two months earlier (November 2019).

-Paragraph from line 140 to 145. They must merge inclusion criteria and then, cite the exclusion criterion (this appears in the middle of the paragraph now, line 142).

Results

-I have missed a summary table or a general summary of all the findings.

-There is a huge difference between the number of articles retrieved and that finally selected, so another exclusion criteria besides non-English language may have been used. They should specify why they discarded so many articles.

-Line 162. According to Table 1, the figures given for articles retrieved and selected for SRH is not correct (13/166). Table 1 shows 19 papers included out of a total of 183.

-The way of citation along the text is not consistent. For instance, the bracket including references 13 and 14 must be placed before the full stop (line 172). The same for reference [18] in line 178, [16] in line 186, [14] and [27] in line 189, etc. Citations are sometimes after, sometimes before full stops.

-Line 177. 820 out of 1254 is more than a half, so they must write it in that way [more than a half reported being unable…].

-Line 190. People who use drugs must be specified that these are injected (intravenous drug users).

-Lines 205-206. They must specify the key population of this study.

-Line 213. The WHO goal to finish the TB pandemic must be referenced with the corresponding citation.

-lines 241 and 242. The sentence referring to losses of jobs should be related to TB. Otherwise, it is not consistent there.

-Lines 324 and 325. They must replace each citation after a country with the citation at the end of the sentence, as they already did before in the previous paragraphs.

-Line 130. I cannot find the relationship between reference 130 and the impact of school closures in children´s nutrition. Is this reference correctly used?

-line 357. Replace “less than five years” by “younger than five years”.

-Lines 359 to 364. They have to insert a citation related to this statement, before writing about maternal mortality.

-Lines 383 and 384. The second part of the sentence (after comma) doesn´t have any reference. Is the reference [136] also suitable? In that case, this reference must be placed at the end of the sentence.

-References

Date of access to webpages must be included at the end of citation (i.e., [9], [138], [145], [146], [160], and so on).

[52], [62]. Pages are missing.

[11]. It is not properly a reference, but a product, a software, without referring to any author. Remove this reference and take the information to the text between brackets. The same as EndNote, reference [12].

[145]. It is not exactly correct how this reference is depicted. Is it a link to a webpage? Must indicate the page and date of access.

-Table 1. I am not sure of “pulled” was the most accurate term. I would rather use “retrieved”. In addition, authors must add an extra column to depict the total amount of each category (included, commentaries, excluded, etc.) and risk condition (HIV, malaria, etc), both published and preprints literature.

7. PLOS authors have the option to publish the peer review history of their article (what does this mean?). If published, this will include your full peer review and any attached files.

Reviewer #1: **Yes: **Okoro Chika Augustus

Reviewer #2: No

---

## [Author Response · Author response to Decision Letter 1]

30 Jul 2022

PONE-D-21-08875R1

Competing Health Risks Associated with the COVID-19 Pandemic and Early Response: A Scoping Review

Due July 18, 2022

• Response to reviewers

• Revised Manuscript with Track Changes

• Manuscript (Clean)

REVIEW COMMENTS TO THE AUTHOR

Reviewer #1: This paper has been properly written. All the previous concerns i had, especially in the presentation of the results were addressed by the authors.

Reviewer #2: 

Abstract

-It is very confusing the figures got from their literature search when mentioned. 1604 published papers and 205 preprints met inclusion criteria, but only 132 and 21 were selected, respectively. They have to replace the verb “including” by “selecting”.

Thank you for this comment. We included articles that had a primary focus on the impact of COVID-19 on one of the five listed public health threats AND if they included empiric data or mathematical models on health or service outcomes. We realize there is some confusion with the phrase “met our inclusion criteria,” and have revised this section of the abstract to the following:

“A total of 1604 published papers and 205 preprints were retrieved in the search. Overall, 8.0% (129/1604) of published studies and 10.2% (21/205) of preprints met the inclusion criteria and were included in this review…”

Introduction

-References are not properly used in some instances. Reference [2] is not related to Public Health responses but to estimating the success of HIV care in key population.

Thank you for catching this. There was an error that occurred when transferring our references between versions. We have now corrected this to include the correct reference:

Hale, T, Angrist, N, Goldszmidt, R, Kira, B, Petherick, A, Phillips, T, et al. A global panel database of pandemic policies (Oxford COVID-19 Government Response Tracker). Nat Hum Behav. doi: 10.1038/s41562-021-01079-8

-Lines 123-124. Delete “rather than systematic review”. If they want to discuss this point, relocate it in Discussion section.

Per your suggestion, we removed “rather than systematic review” from this sentence. We agree that mention of this should be restricted to the Discussion section, which is included in the Limitation sub-section. 

-Methods

-Line 130. The other reviewer has already pointed that this sentence is confusing or trifling. In my opinion, they should merge the first sentence and the following one, or simply delete the first one.

We have revised this sentence to merge the first and the following sentence. 

-Line 131. They must enter the MeSH acronym the first time it was used in the text, as they did it in Abstract.

Thank you for this. We have added what MeSH stands for here. 

-There is a contradictory information regarding period of inclusion. In Abstract, the period cited ranged from January 1st to October 31st, 2020, whereas in Methods (line 144) the period started two months earlier (November 2019).

Thank you for catching this. This should read November 2019 throughout, and we have corrected this.

-Paragraph from line 140 to 145. They must merge inclusion criteria and then, cite the exclusion criterion (this appears in the middle of the paragraph now, line 142).

Thank you for this suggestion. We moved the last sentence indicating the timeline for the review up to be immediately after the other inclusion criteria. The paragraph now reads as follows:

Articles were eligible for inclusion if the primary focus was the impact of COVID-19 on one of the five listed public health threats, and if they included empiric data or mathematical models on health or service outcomes. Articles were included if they were published or posted between November 2019 and October 2020 with the most recent search implemented on October 31, 2020. We excluded articles not written in English. Commentaries were screened for key themes to support interpretation of the measured and modeled data. 

Results

-I have missed a summary table or a general summary of all the findings.

We have presented a summary of findings at the beginning of each health risk section. We did indicate that “COVID-19 has been associated with reduced access to services, decreased diagnosis, and poorer health outcomes for HIV, TB, malaria, and SRH, and increases in malnutrition.” 

We did not include any further summary of the findings across health risks, given concerns about the oversimplification of the results and earlier feedback about repetition across sections of the manuscript.

-There is a huge difference between the number of articles retrieved and that finally selected, so another exclusion criteria besides non-English language may have been used. They should specify why they discarded so many articles.

Thank you for this comment. We noted in the Methods section that the inclusion criteria also stipulated that articles include empiric data or mathematical modeling data. This was the primary factor for determining whether or not articles were included in our review. This requirement has now been clarified, as noted above.

-Line 162. According to Table 1, the figures given for articles retrieved and selected for SRH is not correct (13/166). Table 1 shows 19 papers included out of a total of 183.

Thank you very much for catching this error; we also subsequently noted this error in the abstract and have revised it in both places. 

-The way of citation along the text is not consistent. For instance, the bracket including references 13 and 14 must be placed before the full stop (line 172). The same for reference [18] in line 178, [16] in line 186, [14] and [27] in line 189, etc. Citations are sometimes after, sometimes before full stops.

Thank you for this. This manuscript was the result of the collaborative work of all of the authors using different styles; we have gone through and corrected the inconsistencies. 

-Line 177. 820 out of 1254 is more than a half, so they must write it in that way [more than a half reported being unable…].

We have revised this to specify “more than half…” rather than just “half.”

-Line 190. People who use drugs must be specified that these are injected (intravenous drug users).

Here, we use the phrase “people who use drugs” to include both those who inject drugs and those who do not. Lack of access to services during this period was particularly problematic for those who already had a difficult time engaging in services in the first place, and this included people who used drugs more broadly. 

-Lines 205-206. They must specify the key population of this study.

The study referenced here does not refer to a specific key population, but rather to a population of individuals living with HIV being served by a safety-net HIV clinic in San Francisco. It includes those that are receiving publicly funded insurance and have a higher prevalence of mental illness, substance use, and unstable housing.

-Line 213. The WHO goal to finish the TB pandemic must be referenced with the corresponding citation.

We have revised this sentence and included a citation as appropriate. 

-lines 241 and 242. The sentence referring to losses of jobs should be related to TB. Otherwise, it is not consistent there.

Good point. We have revised this sentence to relate job loss to risk of TB. 

-Lines 324 and 325. They must replace each citation after a country with the citation at the end of the sentence, as they already did before in the previous paragraphs.

We have moved these citations so that they are within the comma for each country, both for consistency across the paper and to clarify any confusion. 

-Line 130. I cannot find the relationship between reference 130 and the impact of school closures in children´s nutrition. Is this reference correctly used?

Thank you. Upon reviewing this further, we realized that this citation (130) was meant to be used in a previous line, indicating that food insecurity could be related to disruption of food supply chains because of limited movement of people and goods and disruption of food markets. We have made this correction.

-line 357. Replace “less than five years” by “younger than five years”.

Thank you; we have made this change. 

-Lines 359 to 364. They have to insert a citation related to this statement, before writing about maternal mortality.

Thank you very much for this careful review of the references. We have added the appropriate citations to this section.

-Lines 383 and 384. The second part of the sentence (after comma) doesn´t have any reference. Is the reference [136] also suitable? In that case, this reference must be placed at the end of the sentence.

Your interpretation is correct; we have moved the reference to the end of the sentence. 

-References

Date of access to webpages must be included at the end of citation (i.e., [9], [138], [145], [146], [160], and so on).

We have gone through all of the citations and added access dates from the following references: 6, 8, 9, 93, 138, 145, 146, 158, 160.

[52], [62]. Pages are missing.

Pages have been added, along with volume and issue numbers. 

[11]. It is not properly a reference, but a product, a software, without referring to any author. Remove this reference and take the information to the text between brackets. The same as EndNote, reference 

[12].

We have consistently cited both EndNote and Covidence as references in prior review papers and feel that it is a reasonable approach for the purposes of a review such as this one. However, we defer to the editors to alert us if the journal has a preference on this point and we would be happy to adjust.

[145]. It is not exactly correct how this reference is depicted. Is it a link to a webpage? Must indicate the page and date of access.

We have added a link to the webpage where the report can be found and added an access date.

-Table 1. I am not sure of “pulled” was the most accurate term. I would rather use “retrieved”. In addition, authors must add an extra column to depict the total amount of each category (included, commentaries, excluded, etc.) and risk condition (HIV, malaria, etc), both published and preprints literature.

Thank you for this. We have changed “pulled” to “retrieved” and added a total column for each risk condition. It would not make sense to add a total column for included, commentaries, excluded as that is reflected in the number retrieved.

---

## [Decision Letter · Decision Letter 2]

9 Aug 2022

Competing Health Risks Associated with the COVID-19 Pandemic and Early Response: A Scoping Review

PONE-D-21-08875R2

Dear Dr. Baral,

We’re pleased to inform you that your manuscript has been judged scientifically suitable for publication and will be formally accepted for publication once it meets all outstanding technical requirements.

Kind regards,

Malik Sallam, M.D., Ph.D.

Academic Editor

PLOS ONE

Additional Editor Comments (optional):

Reviewers' comments:

Reviewer's Responses to Questions

**Comments to the Author**

1. If the authors have adequately addressed your comments raised in a previous round of review and you feel that this manuscript is now acceptable for publication, you may indicate that here to bypass the “Comments to the Author” section, enter your conflict of interest statement in the “Confidential to Editor” section, and submit your "Accept" recommendation.

Reviewer #2: All comments have been addressed

2. Is the manuscript technically sound, and do the data support the conclusions?

Reviewer #2: Yes

3. Has the statistical analysis been performed appropriately and rigorously? 

Reviewer #2: N/A

4. Have the authors made all data underlying the findings in their manuscript fully available?

Reviewer #2: Yes

5. Is the manuscript presented in an intelligible fashion and written in standard English?

Reviewer #2: Yes

6. Review Comments to the Author

Reviewer #2: Thank you for addressing all my suggestions and amending the mistakes found in the first round. I agree with the publication of this manuscript as the current reviewed version.

7. PLOS authors have the option to publish the peer review history of their article (what does this mean?). If published, this will include your full peer review and any attached files.

Reviewer #2: **Yes: **González-Domenech, CM

---

## [Editor Report · Acceptance letter]

18 Aug 2022

PONE-D-21-08875R2 

Competing Health Risks Associated with the COVID-19 Pandemic and Early Response: A Scoping Review 

Dear Dr. Baral:

I'm pleased to inform you that your manuscript has been deemed suitable for publication in PLOS ONE. Congratulations! Your manuscript is now with our production department. 

Kind regards, 

on behalf of

Dr. Malik Sallam 

Academic Editor

PLOS ONE